# Ligand-induced conformational selection predicts the selectivity of cysteine protease inhibitors

Geraldo Rodrigues Sartori[1], Andrei Leitão[1], Carlos A. Montanari[1], Charles A. Laughton[2]*

**1** Grupo de Química Medicinal do IQSC/USP, Instituto de Química de São Carlos, Universidade de São Paulo, São Carlos, São Paulo, Brazil, **2** School of Pharmacy and Centre for Biomolecular Sciences, University of Nottingham, Nottingham, England, United Kingdom

* Charles.laughton@nottingham.ac.uk

**Data Availability Statement:** Data can be found at: https://zenodo.org/record/3518308 - 10.5281/zenodo.3518308; https://zenodo.org/record/3522090 - 10.5281/zenodo.3522090; https://

## Abstract

Cruzain, a cysteine protease of *Trypanosoma cruzi*, is a validated target for the treatment of Chagas disease. Due to its high similarity in three-dimensional structure with human cathepsins and their sequence identity above 70% in the active site regions, identifying potent but selective cruzain inhibitors with low side effects on the host organism represents a significant challenge. Here a panel of nitrile ligands with varying potencies against cathepsin K, cathepsin L and cruzain, are studied by molecular dynamics simulations as both non-covalent and covalent complexes. Principal component analysis (PCA), identifies and quantifies patterns of ligand-induced conformational selection that enable the construction of a decision tree which can predict with high confidence a low-nanomolar inhibitor of each of three proteins, and determine the selectivity for one against others.

## Introduction

Cysteine proteases, specifically those belonging to the Papain family [1](family C1 and Clan CA, in the MEROPS [2] database classification), can be found in cellular lysosomes and the extracellular matrix after cellular secretion [3,4], and are involved in many different biological processes and pathways, including development and growth, cellular signalling, apoptosis, pro-hormone processing, nutrition, invasion of host cells and others [5]. Unsurprisingly then, they are of significant interest to the pharmaceutical industry[6,7] and amongst these enzymes are validated targets for the development of treatments for osteoporosis, cancer, immune diseases and parasitic diseases[8] (e.g. malaria, Chagas disease [9], and leishmaniasis). As a result of this interest there are currently a considerable number of small molecule inhibitors of cysteine proteases, especially of human cathepsins, [10–12] *Trypanosoma cruzi* cruzain, [13–15] and falcipains from *Plasmodium falciparum* [16,17].

Chagas Disease is a parasitic disease caused by the flagellated parasite *Trypanosoma cruzi* and was described for the first time in 1909 by Carlos Chagas [18–20]. Despite the high economic cost of Chagas disease, estimated at 7 billion dollars per year [21] due to palliative

zenodo.org/record/3523307 - 10.5281/zenodo.
3523307; https://zenodo.org/record/3523367 - 10.
5281/zenodo.3523367.

**Funding:** AL, CAM and CAL thank the Conselho
Nacional de Desenvolvimento Científico e
Tecnológico – CNPq (grant #400658-2014-3,
http://www.cnpq.br) and Fundação de Amparo à
Pesquisa do Estado de São Paulo – FAPESP (grant
#2013/18009-4, http://www.fapesp.br) for
financing this project. This work used the ARCHER
UK National Supercomputing Service (http://www.
archer.ac.uk), with additional access through
HECBioSim (EPSRC Grant EP/L000253/1, http://
www.epsrc.ac.uk). The funders had no role in
study design, data collection and analysis, decision
to publish, or preparation of the manuscript.

**Competing interests:** The authors have declared
that no competing interests exist.

treatment and early retirement, this disease is neglected by the pharmaceutical industry. The
current available treatment is the drug benzonidazole, which was developed during the 1970s
and has severe side effects [22]. The *T. cruzi* enzyme cruzipain (Enzyme Classification number
3.4.22.51) is abundant throughout the life cycle of the parasite and is particularly important
during the amastigote phase. Cruzipain is essential to parasite nutrition as well as during dif-
ferentiation phases and host cell invasion, when it activates inflammatory process and
degrades immunoglobulins. It is expressed as a zymogen consisting of catalytic, pre and pro
domains. The last is cleaved to obtain the mature enzyme, while the pre-domain, highly glyco-
sylated, is maintained as part of the enzyme and only released later to trigger an immunogenic
response in the host organism [23]. The presence of this domain in cruzipain differentiates it
from human cathepsins [24]. However, it has been shown that the cleavage of the pre-domain
does not affect the catalytic activity of the protein [25] and hence most research focuses on the
N-terminal catalytic domain, which when expressed heterologously in Escherichia coli is
named cruzain. Cruzain is composed of 215 residues of which Cys25 and His159 are the cata-
lytic dyad. It features three disulfide bonds distributed over the protein that enhance extracel-
lular stability and are also present in cathepsin enzymes.

Cysteine proteases are effectively inhibited by several classes of covalent and non-covalent
inhibitors [14, 26]. Covalent inhibitors typically present groups containing a reactive electro-
philic centre (warhead) susceptible to nucleophilic attack by the activated cysteine present in
the enzyme active site. Depending on the nature of the warhead, covalent bond formation
between the inhibitor and the protein may be irreversible or reversible. [27]. Examples of
reversible covalent inhibitors are peptide-aldehydes, α-diketones, α-ketoesters, α-ketoamides,
α-keto acids and nitriles [28]. In contrast, compounds such as peptidyl diazomethyl ketones,
fluoromethyl ketones, epoxides and vinyl sulfones are capable of binding irreversibly to the
catalytic cysteine, acting as "suicide" inhibitors [29]. Several non-covalent inhibitors of prote-
ases have also been described in the literature, including the analysis of the mode of binding of
our cruzain inhibitor Neq0176 [14]. However, few of them exhibit comparable enzyme affinity
to those acting via a covalent mechanism [30], with the notable exception of a new (non-cova-
lent) and reversible competitive inhibitor that inhibits cruzain in nanomolar concentration
and another one that also kills *T. cruzi* in low micromolar range [31].

The rational design of selective, potent reversible covalent inhibitors of cysteine proteases is
complicated by their mechanism of action. Aspects of both their covalent and non-covalent
interactions must be considered, but how to do so remains an open question. We have recently
shown [32] that careful QM/MM calculations on the covalent binding of dipepdidyl nitrile
ligands to cruzain can correlate well with experimental binding data, but the procedure is not
suitable for high throughput use where the interactions of multiple ligands with multiple
enzymes requires to be evaluated. Recently, Waldner et al. [33] have shown that, for a panel of
serine proteases, substrate specificity that could not be understood from an analysis of static
(crystal) structures, was interpretable once the differential dynamics of the proteins was con-
sidered. It is well established that any attempt to understand and optimise a ligand-protein
interaction must take into account protein flexibility [34, 35]. The two limiting models for
ligand-induced conformational change in a protein are induced fit and conformational selec-
tion [36]. The former supposes that the native, free, form of protein has a preference for one or
more specific conformations with which the ligand interacts, which then induces and stabilize
a new conformation of protein that was not accessed originally [37]. The latter hypothesises
that the normal thermally-activated dynamics of the free protein involves it spontaneously but
transiently adopting the conformation appropriate for ligand binding. In the presence of the
ligand this state is captured and 'titrated out' of the equilibrium distribution [38]. These two
limiting models make very different predictions about the effect of ligand binding on the

conformational dynamics of the protein, and thus configurational entropy components of the binding free energy. The induced fit model involves the ligand encouraging the protein to visit conformational states it did not sample in the *apo* state–configurational entropy may tend to increase. In contrast, the conformational selection model involves the ligand discouraging the protein from adopting certain conformations, so configurational entropy may tend to decrease. Molecular simulation methods can provide powerful insights into this process, [39, 40], and so rationally guide ligand design.

Here we use molecular dynamics simulations to investigate the effects of a range of nitrile-based cysteine protease inhibitors, with affinities ranging from the micromolar to nanomolar range, on the conformational dynamics of the active sites of cathepsin K, cathepsin L, and cruzain. We hypothesised that analysis of the ways in which different ligands perturb and limit the intrinsic dynamics of the *apo* enzymes might give insights into the origins of affinity and selectivity, and indeed this proves to be the case. Our approach is somewhat different from that described by Waldner et al. but the outcomes are similar: with the aid of principal component analysis (PCA), we identify and quantify patterns of ligand-induced conformational selection that enable the construction of a decision tree which can predict with high confidence a low-nanomolar inhibitor of each of three proteins and determine the selectivity for one against others.

## Methods

### Ligand and protein data set

To evaluate the capacity of molecular dynamics simulations to discriminate between inhibitors of cruzain, cathepsin K and cathepsin L, four ligands with known potency against all three proteins were chosen from the literature. These ligands were selective for cruzain over cathepsin L (compound 26 from Beaulieu et al. [13], here called "ICR"), cathepsin K over cathepsin L (odanacatib [41], "ICK"), cathepsin L over cathepsin K (compound 17 from Asaad et al. [42], "ICL") and cathepsin K over cruzain (compound 8 from Black et al. [43], "IKR"). A fifth ligand, BCR, was added to the data set so it included at least one molecule known to be inactive against cruzain. (Fig 1 and Table 1). To test the decision tree that this work ultimately produced, we additionally selected four ligands from our own laboratory: Neq0409, Neq0544, Neq0538, and Neq0539, that have known Ki against cruzain. The protein crystal structures used were 2OZ2 (cruzain [44]), IMEM (cathepsin K, [45]) and 2YJ2 (cathepsin L [46]).

### Docking studies

Protein and ligand preparation and docking was done using tools from the Schrodinger software suite [49]. For the proteins, all non-protein atoms were removed and disulfide bond and missing sidechain atoms added using the *ProteinPreparationWizard* tool. Protonation states of residues were calculated at pH 5.8 using the PropKa algorithm implemented in *Maestro*. The catalytical diad Cys25 and His159 in cruzain (and the equivalents in the cathepsins) were considered as deprotonated and protonated, respectively. Also, cruzain Asp57 was defined as protonated. A default minimization implemented on *ProteinPreparationWizard* tool was performed to relax the systems.

The ligand K777 (complexed with cruzain in PDB code 2OZ2) was used as the reference molecule to define a box of 15 Å from its centroid for the docking procedure. Furthermore, it was also used to define three hydrogen bond constraints: between the ligand and the backbone N and O atoms of G66, and between the ligand and the backbone O atom of D158 (cruzain numbering). During the docking procedure, using the *StandardPrecision* parameters, poses were only selected if they formed at least one hydrogen bond with any of these atoms. *Glide*

**Fig 1. Structure of dipeptidyl nitriles used in this work.** Top, left to right: ICR, ICK, ICL. Centre, left to right: IKR and BCR. Bottom, left to right: Neq0409, Neq0544, Neq0568, Neq0569.

was permitted to return a maximum of 10 different poses; other parameters were set at default values. The ligands were prepared using *LigPrep* for pH 5.8, the same as used in cruzain enzymatic assays [14].

## Molecular dynamics simulations

**Ligand parameterisation.** All simulations were performed using *Amber14*. Electrostatic potentials for each ligand (formal charge predicted by LigPrep for a pH of 5.8) were calculated at the HF/6-31G+ level using Gaussian09 [50]. No geometry optimization was done, so as to preserve the docked pose of ligand. Then, RESP [51] partial atomic charges were derived using R.E.D-vIII.4 [52] and Ante-RED-1.4 scripts. The Antechamber package implemented in AmberTools14 was used to generate the gaff (general amber force field) [53] library and topology parameters for each ligand. For covalent ligands, we initially generated a molecule consisting of the studied ligand covalent bounded to a cysteine residue, which has its N and C

**Table 1. IC50 values, in nM, of dipeptidyl nitriles against cruzain, cathepsin K and cathepsin L.** Values in italics are Ki values rather than IC50s.

| Ligand | Cruzain | Cathepsin K | Cathepsin L | Ref |
|--------|---------|-------------|-------------|-----|
| ICR | 0.4 | 2 | 1060 | [13] |
| ICK | *40* | 0.2 | 2995 | [41] |
| ICL | - | >10000 | 20 | [42] |
| IKR | 1,8 | 0.005 | 47 | [13],[47] |
| BCR | *6300* | - | - | [48] |
| Neq040 | *453* | - | - | [48] |
| Neq054 | *159* | - | - | [48] |
| Neq056 | *56* | - | - | [48] |
| Neq056 | *25* | - | - | [48] |

terminal capped by ACE (acetyl) and NME (methyl amine) groups respectively. The ligand portion of the cross-linked molecule was named using gaff nomenclature while cysteine atoms were named using FFF14SB nomenclature. All steps were done as previously described, except for the RESP derivation step. On this procedure, the N, H, C and O atoms had charges fixed to -0.4157, 0.2719, 0.5973 and -0.5679, respectively (as per standard AMBER amino acids). In addition, the caps ACE and NME were defined to have final charge zero and to be removed from final molecule. After the charge derivation, a connectivity table was generated using Antechamber and the missing parameters due to the interfaced gaff-FF14SB nomenclature were estimated from comparable connections in the gaff force field. The C and N terminal of the non-natural residue was defined in tleap as being the C and N atom of the cysteine.

**Protein preparation.** The protein structure created for the docking procedure was used as the starting point to prepare the molecular dynamics complex. Initially, all hydrogen atoms were removed to avoid nomenclature incompatibility. The disulfide cysteines were renamed to CYX, and the catalytic C25, to CYM. Histidine residues were renamed as HIP, HID or HIE according to it tautomer/protonation state. The only CONECT lines kept from the .pdb file were those relating to disulfide bonds. Noncovalent ligand information was included at the end of file for complexes where covalently cross-linked adducts replaced the standard C25 residue, with the same nomenclature used in the library file previously prepared.

**Parameterisation of protein-ligand complexes.** Once the .pdb file of each complex (covalent or non-covalent) was prepared and all the ligands were parametrised, prmtop (topology file) and inpcrd (initial coordinate files) for each complex were generated based on FF14SB (protein atoms) and gaff (ligand atoms) force fields using a tleap script. Systems were neutralised by the addiction of Na+ or Cl- ions and then solvated with TIP3P [54] water in a truncated octahedral box extending at least 10 Å beyond any protein atom.

**Equilibration.** Prior to production MD, the equilibration process started with 2000 cycles of minimization of solvent and ions. A position restraint of 500 kcal.mol$^{-1}$ was applied to protein and ligand atoms. After that, the whole system was minimized for 3500 cycles without any restraints. All MD simulations were done using a cut-off of 10 Å for non-bonded interactions and a time step of 2 fs, SHAKE [55] was used to constrain bonds to hydrogen atoms and the Langevin approach to control the temperature (collision frequency 2 s$^{-1}$). First, the system was heated to 310 K over 50ps using the NVT ensemble and a position restraint of 10 kcal.mol$^{-1}$ on protein and ligand atoms. After the heating procedure, a density equilibration at NTP were carried out over 300 ps followed by an extended equilibration phase of 5 ns. The 100 ns production simulations were run in the NTP ensemble in 5 ns "chunks", with a new seed for Langevin dynamics to restart each simulation. Coordinates were saved every 50ps.

**Data selection and collection.** To minimize artefacts that might be due to imperfect construction of the initial systems, we adopted a two-stage simulation strategy. First (round 1) we ran five replicate 100ns simulations on each complex, with randomized initial velocities. The final structure from the replicate that showed the greatest conformational stability over the first 100ns was then used as the start point for five further 100ns simulations (round 2), again each beginning with a new randomized set of velocities. In cases where all five replicates in round 1 showed stable behaviour, round 2 was not performed.

## Molecular dynamics simulation analysis

Data regarding the RMSD of the ligand and the distance between the ligand nitrile carbon and the sulfur of C25 (distance C-N) data were calculated for production simulations using *cpptraj* as implemented within the *AmberTools14* package. Frame alignment and RMSD calculation used the docked conformation as the reference. Graphs were generated using *xmgrace*.

Principal component analysis, was performed using *PyPcazip* [56]. The calculations were done using default parameters. One PCA was performed for each of the ligands studied. Each PCA included all relevant simulation data, taking a sample every 100ps. Thus, for each of ligands ICR, ICK, ICL, and IKR, the PCA included: a) the data from the five replicate simulations of cruzain, cathepsin K and cathepsin L in apo form; b) the data from the simulations of the same three proteins when non-covalently complexed to the ligand; and c) the data from the simulations of the same three proteins when covalently cross-linked to the ligand. In the case of ligand BCR, since we only have binding data for the interaction of this ligand with cruzain, the dataset for PCA contained just apo-cruzain, plus the non-covalent and covalent complexes of BCR with cruzain.

To permit a full comparison of PCA results, a common subset of atoms was selected from every trajectory as follows. Firstly, all residues in the reference cruzain-K777 complex (PDB:2OZ2), that had at least one atom within 7 Å of the ligand were selected. Then by structural alignment the corresponding residues in cathepsin K (PDB code 1MEM) and cathepsin L (PDB: 2YJ2) were identified. At positions where the amino acid differed between one protein and another, the maximum common subset of structurally equivalent atoms was retained.

### Conformational distribution analysis

For each ligand, we analysed the PCA data to observe, in the PC1/PC2 plane, the conformational space sampled by the common subset of protein binding site atoms in the *apo*, non-covalently bound, and covalently bound complexes. We found that for every ligand, the sampling of the PC1/PC2 plane lay within a bounding box of -20–20 Å in each dimension. Therefore we calculated 2D histograms for each dataset within these limits, with a 1 Å$^2$ resolution (thus 20 x 20 bins). For each ligand we then defined three (potentially overlapping) sets. The first, **A**, contains all bins that are sampled by the *apo* protein. The second, **N**, similarly contains all bins sampled by the protein when non-covalently bound to the ligand. The third, **C**, contains all bins sampled in the simulation of the relevant covalent complex. For all sets, bins with less than 1% occupancy were ignored. Using standard set theory notation, the number of samples in **A** is |**A**|, etc., the set of bins sampled in both the *apo-* and non-covalent simulations is **A**∩**N**, etc., and the number of samples in this intersection is |**A**∩**N**|, etc.

## Results

### Structural analysis of molecular dynamics simulations

Our 2-round, 5-replicate, MD protocol was designed to maximize the chance of obtaining a "well-behaved" trajectory data set for each non-covalent ligand simulation. We expected that simulations of ligand complexes that were known to have a high binding affinity would be more likely to be stable than those where it was known the ligand was a poor inhibitor of the protein, and in general this was what was observed. The two quality metrics were a) the distance between the S atom of the catalytic cysteine and the C atom of the warhead nitrile; and b) the RMSD of the ligand from its original, docked, conformation. The results obtained are summarized in Table 2 and in S1–S5 Figs. For ligands IKR, ICR, and ICL, there appears some relationship between these metrics and binding affinity. However for ICK the simulations suggest a poor geometry for the non-covalent complex with cruzain, and a poor warhead geometry for the complex with CatK, despite the fact it is a good (or reasonable) inhibitor of both enzymes. Conversely, the complex of BCR with cruzain appears to have a good stability and geometry but is known to be a poor inhibitor. In this last case, close inspection of the binding pose shows that though the nitrile-cysteine distance is not long, the relative orientations of the groups are not conducive to nucleophilic attack, so this may explain this anomaly. In

**Table 2. Summary of metrics extracted from the MD simulations of the non-covalent ligand-protein complexes: Distance C-N (distance between thiolate of Cys25 and nitrile carbon of ligand, in Ångstroms) and RMSD of ligands from the initial docked conformation (in Ångstroms) during the second round of simulations.** Cells are colour coded according to experimentally determined inhibitory activity, orange: < 5 nM; tan: < 50 nM; red: > 1000 nM; white: not determined.

| Protein Ligand | Distance C-N | | | RMSD | | |
|---|---|---|---|---|---|---|
| | cruzain | cathepsin K | cathepsin L | cruzain | cathepsin K | cathepsin L |
| ICL | 4.9 | 5.1 | 4.7 | 1.9 | 4.9 | 2.0 |
| IKR | 6.4 | 4.1 | 8.0 | 6,6 | 2.5 | 5.9 |
| ICR | 5.1 | 3.4 | 6.1 | 5.5 | 3.6 | 5.7 |
| ICK | 10.0 | 7.7 | 6.1 | 6.8 | 3.1 | 9.2 |
| BCR | 5.4 | - | - | 5.4 | - | - |

summary, neither of these simple quality metrics provides a reliable guide to ligand potency/selectivity.

## Binding free energy calculations using the MMGBSA method

Moving beyond simple structural metrics, the relative binding free energy of complexes was estimated using the MMGBSA approach. Since these nitrile inhibitors are covalent-reversible inhibitors of these enzymes, the application of this approach requires some justification. The chemistry of the cross-linking reaction is the same for all the ligands and all the enzymes studied here–nucleophilic attack of the protein cysteine sulfur atom at the ligand nitrile carbon, with the formation of a thioimidate adduct. Thus, while the energetics of this process are likely to be a major contribution to the overall thermodynamics of ligand binding, they are unlikely to be the source of observed patterns of selectivity/affinity. The assumption of the MMGBSA approach as used here is that this term is in effect a constant, so that the other terms the calculation takes into account–desolvation terms and non-covalent interactions–should correlate with observed binding affinity. However this did not prove to be the case; the method did not satisfactorily differentiate between the low nanomolar and low micromolar complexes (see S7 and S8 Figs). Rather than put this down to our assumption that the covalent term is more or less constant, we hypothesized that part of the reason for the poor performance of the MMGBSA method in this case could be due to incomplete consideration of the entropic term, particularly ligand-induced changes in the flexibility of the proteins. It is well known that the conventional MMGBSA approach does not handle such terms well, so we decided to investigate this aspect in detail.

## Principal component analysis of cysteine protease active site conformations in presence of noncovalent and covalent ligands

Using the meta-trajectory PCA approach described in the methods section, we analysed, for each protein, how the conformational space sampled by the active site region was perturbed (shifted, and/or constricted) by both non-covalent, and then covalent, binding of each ligand. Results are presented in Fig 2, and in more detail in S9–S13 Figs. In each case the PC1/PC2 space sampled by the apo protein is indicated by the black dots, by red dots for when the protein is bound non-covalently to the ligand, and by green dots when there is a covalent complex. Broadly speaking, we would expect conformational selection-type behaviour to be manifested by a shrinkage in the sampling within the boundaries of the previous state, while induced fit-type behaviour would lead to a shift in the distribution so that previously unsampled states became populated, and previously populated ones not.

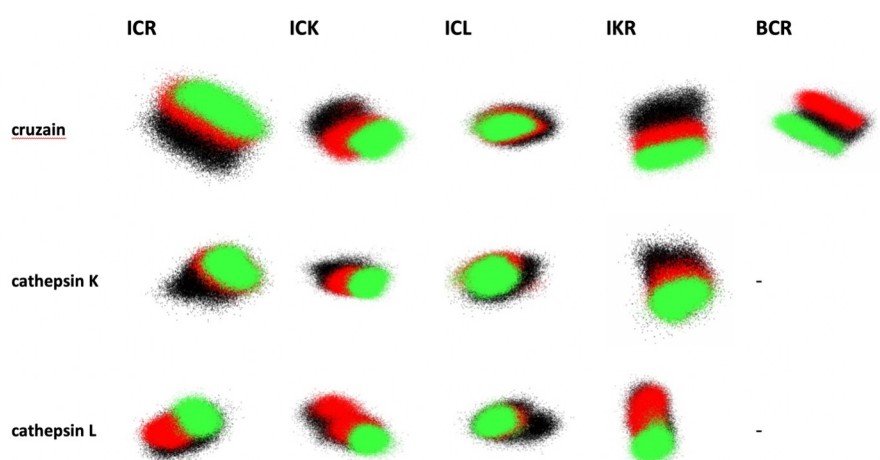

**Fig 2. Projections onto the first two principal components of cruzain (first row), cathepsin K (second row) and cathepsin L (third row) simulations in their apo form and complexed with noncovalent and covalent forms (black, red and green dots, respectively) of ligands ICR, ICK, ICL IKR and BCR.** Each panel is at the same scale (40 Ångstroms in both X and Y).

Our results can be divided into complexes with activity below and above 500 nM, respectively. The first class is composed of the complexes ICR-cruzain, ICR-cathepsin K, ICK-cruzain, ICK-cathepsin K, ICL-cathepsin L, IKR-cruzain, IKR-cathepsin K, IKR-cathepsin L. In this class, we observe a common pattern for the changes in the conformational space sampled by the active site of the enzymes as we transit from *apo* form to noncovalent and then covalent complexes (black, red and green dots respectively). This is characterized by a gradual reduction of flexibility of the protein active site. The black dots cover a largest space and the red dots are contained into this space. Furthermore, the green dots are in turn contained into noncovalent space delimited by red dots. The fact that this behaviour was observed for all three proteins allows us to make a well-substantiated proposal that these cysteine proteases adopt the conformational selection model of flexibility when complexed with low nanomolar dipeptidyl nitrile inhibitors.

In contrast, the second class (ICL-cathepsin K, ICK-cathepsin L, ICR-cathepsin L, IKR-cathepsin L, BCR-cruzain) does not present a common pattern for different complexes with respect to the conformational space covered by protein active site in each kind of simulation. This second class can be subdivided into three different subclasses according to the behaviour of protein active site in presence of noncovalent and covalent form of ligand.

The first subclass is comprised by the complexes ICR-cathepsin L, IKR-cathepsin L, and BCR-cruzain, in which the noncovalent form of ligand stabilizes a distinct conformation of protein active site when compared with conformations accessed by this site in (the maybe only theoretical) complex with the corresponding covalent ligand. In these cases, we propose that low inhibitory activity is due to the fact that the protein must undergo two different conformational changes (apo → noncovalent complex, then noncovalent complex → covalent complex), and the noncovalent ligand is not able to stabilize a conformation in the active site in which the nucleophilic attack is favourable.

The second subclass is represented by complex ICK-cathepsin L, where the conformational space of protein in presence of noncovalent form of ligand is almost the same of the apo form, meaning that protein does not appear to 'feel' the presence of the ligand. The weakness of the ICK-cathepsin L interaction is also evident from the observation that of all the studied complexes, this is the one with the highest ligand RMSD, indicative of the ligand tending to leave the active site over the period of the MD simulations.

It is interesting to observe that the behaviour of this ligand in complex with both cruzain and cathepsin L protein is almost the same, but the PCA analysis here shows two different patterns. The cruzain-ICK complex shows two sequential reductions in conformational space covered by simulation, the first after noncovalent ligand binding and the second one after the formation of the covalent bond, suggesting the ligand induces a conformational selection from all possible states of the protein to a specific one that permits a better interaction with the ligand. In contrast, the conformational spaces covered by cathepsin L in both the free state and in noncovalent complex with ICK are almost the same, suggesting the protein does not properly recognize the ligand. In the (maybe hypothetical) covalent complex, the enzyme is forced to adapt to ligand because of the imposition of bond formation between Cys25 and ligand nitrile.

Lastly, in the case of the ICL–cathepsin K system we see that, remarkably, the conformational space accessed by the noncovalent complex smaller than that for the covalent complex, what may suggest the covalent form ligand induces the protein to a less stable conformation in comparison with noncovalent complex, maybe due to some steric repulsion.

## Development of a decision tree to identify strong and weak complexes

We hypothesised that it might be possible to convert our observations regarding the effects of strong and weak ligands on the dynamics of cysteine protease active sites into a predictive tool. From the parameters obtained by application of set theory to the PCA data (see methods section and S1 Table), a decision tree to identify strong and weak complexes (pK$_i$ or pIC$_{50}$ above and below 7, respectively) was built (Fig 3) that classified correctly all twelve complexes from training set.

The main metric of the decision tree, ($|A \cap N \cap C|)/(|C|$), measures the fraction of conformational space of protein active site in the covalent complex that is accessible in all three states (apo, non-covalently bound, and covalently-bound). A high value for this parameter means the covalent form of ligand only restricts the conformational space of apo and noncovalent state of protein active site without the emergence of a new conformation, i.e. it fits the conformational selection model of flexibility. In fact, a value for this parameter above 60% is found for ICR-cruzain, ICR-cathepsin K, ICK-cruzain, ICK-cathepsin K, ICL-cathepsin L and IKR-cathepsin K, all of which are considered strong complexes. On the other hand, if the protein-ligand pair presents a low overlap between the conformational space covered by each three states ($|A \cap N \cap C|)/(|C| < 60$%) and a high overlap between the conformational space covered by apo and covalent system ($|A \cap C|)/(|C| > 80$%) this implies a large conformational change occurs in the protein active site between the non-covalent and covalent states. The three complexes that presented this kind of behaviour: ICK-cathepsin L, ICR-cathepsin L, and BCR-cruzain, are all examples of weak inhibitors.

The last parameter chosen, for the third branch of the decision tree, was ($|N \cap C|)/(|N|$). A high value for this parameter indicates (if account is taken of the partitioning of examples higher up in the decision tree) that the noncovalent form of ligand may be forcing the protein to a conformation not accessed by apo form. This would be an indicative of induced-fit model for the system. In our case, for cysteine proteases in complex with dipeptidyl nitriles, a value

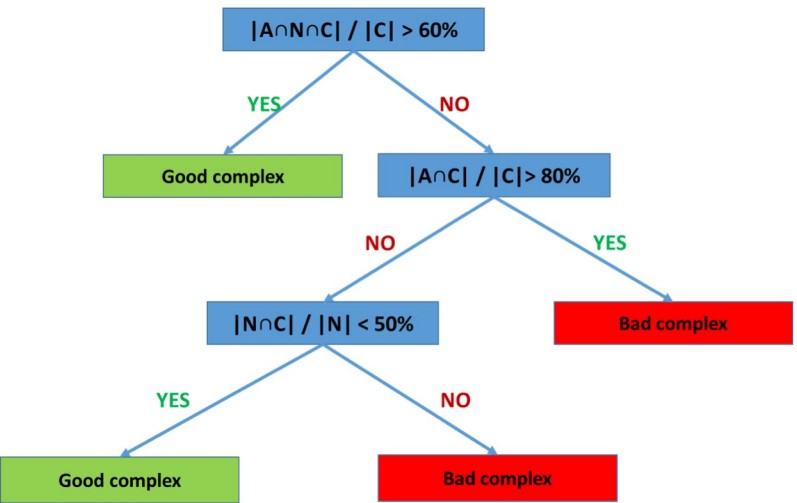

**Fig 3. Decision tree for identification of low nanomolar cysteine protease–dipeptydil nitrile complexes on the basis of conformational space overlaps.**

above 50% for this rule identifies the ICL-cathepsin K complex, which is defined as a weak complex. This result strengthens the hypothesis that conformational selection behaviour for ligands of cysteine proteases is a mark of strong inhibitors, though more examples of complexes that follow this apparent induced-fit behaviour are required to support a general conclusion that this correlates with poor binding.

## Test of the decision tree

To test the decision tree, we applied it to four ligands from our in-house collection of dipeptidyl nitriles (Fig 1 and Table 1). Two of these are potent inhibitors of cruzain, one is moderate (but on the basis of our <500nM cut-off, is classified as active), and one is poor. Simulations of these ligands bound both non-covalently and covalently to cruzain were conducted using exactly the same protocol as described above, and the data analysed in the same way. The decision tree predicts Neq0409-cruzain complex as active at the first node, as $(|A \cap N \cap C|)/|C|$ for this system is 71%. The other three cruzain complexes, with Neq0544, Neq0568 and Neq0569, pass through to the third node, as they have values of $(|A \cap N \cap C|)/|C|$ of 58%, 38% and 33% respectively, and values for $(|A \cap C|)/|C|$ of 75%, 63% and 38% respectively. At this node, $(|N \cap C|)/|N|$ of 52%, 37% and 26% classifies Neq0544, Neq0568 and Neq0569 as inactive, active and active, respectively. In this way the potencies of all four test molecules against cruzain are also successfully classified by the decision tree.

## Conclusions

In this work, we demonstrate the use of molecular dynamics simulations to identify low nanomolar complexes of dipeptidyl nitriles and cysteine protease by the analysis of the conformational space accessed by protein active site during simulations of free form of protein and noncovalent and covalent complexes.

We predict that these cysteine proteases make use of the conformational selection model for ligand selectivity. We observe that the active site in its free form accesses an extended conformational space that is reduced stepwise by the noncovalent, then covalent, binding of ligands. Besides the space reduction, we observe the covalent space lies within the noncovalent space which in turn lies within the free protein space, a typical behaviour of proteins that obey the conformational selection model. Furthermore, this pattern is only observed for complexes with IC50 or Ki below 500 nM. For weak complexes, the conformational space sampled by protein active site in covalent simulations is not a subset of the noncovalent space. This mode of analysis has predictive power: a set theory treatment of PCA distributions can generate a decision tree capable of categorizing the activity of different dipeptidyl nitriles against at least papain-family proteins.

Simpler metrics based purely on analysis of the non-covalent complexes, e.g. RMSD of the ligand from the conformation suitable for cross-linking, or distance between the key cysteine S and nitrile C atoms, are not reliable. At a higher level, MMGBSA binding energy calculations are also not predictive. In part this is not unexpected, since these approaches do not deal in any detail with the process of transforming from the non-covalent to the covalent complex. What our studies reveal is that this step has energetics that do not necessarily correlate with those of the previous step. Though considerably more compute intensive that simple non-covalent docking, the approach described here is has the potential to be applied in a medium-throughput way more realistically than, for example, QM/MM investigations of the cross-linking reaction. We hypothesise that the success of our approach is due to the fact that it captures most of the sources of energetic variation in the dataset–the variation in the structure of the ligands, and the variation in the structure of the receptors. The only terms that is ignored is the actual energy of cross-link formation, and since all ligands and receptors feature the same reactive groups, this is more or less constant. Of course, if in the future we wished to extend the approach to other classes of ligands with different warheads, some modification of the approach would be required as the assumption of a constant cross-link energy term would not hold. One could imagine adding a QM-derived term to the model to account for this. Note that as the chemistry of cross-linking would be the same for all target proteins, we would only need to calculate this term once, and potentially using a small model system.

The workflow used here could be readily automated and should provide a useful tool for the optimisation of high affinity cruzain inhibitors that avoid cross-reactivity with host cysteine proteases. The approach is also quite generic and could be explored for any drug design project that wishes to exploit the covalent modification of the target.

## Supporting information

**S1 Table. Auxiliary metrics derived from two first principal components obtained for system all systems studied.** A, N and C are defined as the set of bins occupied by Apo, noncovalent and covalent simulations and ∩ the intersection of set.
(PDF)

**S1 Fig. Distance between ICR nitrile and sulfur from Cys25 residue (first column) and RMSD of IKR ligand (second column) complexed with cruzain, cathepsin K and cathepsin L (first, second and third row respectively).** Black vertical bars delimit the replicates. Black and red lines represent respectively Round 1 and 2 simulations.
(PDF)

**S2 Fig. Distance between ICK nitrile and sulfur from Cys25 residue (first column) and RMSD of ICK ligand (second column) complexed with cruzain, cathepsin K and cathepsin**

**L (first, second and third row respectively).** Black vertical bars delimit the replicates. Black and red lines represent respectively Round 1 and 2 simulations.
(PDF)

**S3 Fig. Distance between ICL nitrile and sulfur from Cys25 residue (first column) and RMSD of ICL ligand (second column) complexed with cruzain, cathepsin K and cathepsin L (first, second and third row respectively).** Black vertical bars delimit the replicates. Black and red lines represent respectively Round 1 and 2 simulations.
(PDF)

**S4 Fig. Distance between IKR nitrile and sulfur from Cys25 residue (first column) and RMSD of IKR ligand (second column) complexed with cruzain, cathepsin K and cathepsin L (first, second and third row respectively).** Black vertical bars delimit the replicates. Black and red lines represent respectively Round 1 and 2 simulations.
(PDF)

**S5 Fig. Distance between BCR nitrile and sulfur from Cys25 residue (first column) and RMSD of the ligand (second column) complexed with cruzain.** Black vertical bars delimit the replicates. Black and red lines represents respectively Round 1 and 2 simulations.
(PDF)

**S6 Fig. Distance (first column) between ligand nitrile and sulfur from Cys25 residue and RMSD of ligand (second column) complexed with cruzain.** Black vertical bars delimit the replicates. Black and red lines represent respectively Round 1 and 2 simulations. The rows are respectively Neq0409, Neq0544, Neq0569, Neq0568.
(PDF)

**S7 Fig. Binding free energy over the time of round 1 of simulations for ligands ICR, ICK, ICL and IKR (first, second, third and fourth column respectively) complexed with Cruzain (first row), Cathepsin K (second row) and Cathepsin L (third row).** Different colors represented different replicates of the same system.
(PDF)

**S8 Fig. Binding free energy over the time of round 2 of simulations for ligands ICR, ICK, ICL and IKR (first, second, third and fourth column respectively) complexed with cruzain (first row), cathepsin K (second row) and cathepsin L (third row).** Different colors represented different replicates of the same system.
(PDF)

**S9 Fig. Projection over the first two principal components of cruzain (first row), cathepsin K (second row) and cathepsin L (third row) simulation frames in it apo form and complexed with noncovalent and covalent forms of ligand ICR (black, red and green dots, respectively).**
(PDF)

**S10 Fig. Projection over the first two principal components of cruzain (first row), cathepsin K (second row) and cathepsin L (third row) simulation frames in it apo form and complexed with noncovalent and covalent forms of ligand ICK (black, red and green dots, respectively).**
(PDF)

**S11 Fig. Projection over the first two principal components of cruzain (first row), cathepsin K (second row) and cathepsin L (third row) simulation frames in it apo form and**

complexed with noncovalent and covalent forms of ligand ICL (black, red and green dots, respectively).
(PDF)

**S12 Fig. Projection over the first two principal components of cruzain (first row), cathepsin K (second row) and cathepsin L (third row) simulation frames in it apo form and complexed with noncovalent and covalent forms of ligand IKR (black, red and green dots, respectively).**
(PDF)

**S13 Fig. Projection over the first two principal components of cruzain simulation frames in it apo (black dots) form and complexed with noncovalent (red dots) and covalent forms (green dots) of ligands Neq0409 (first row), Neq0544 (second row), Neq0569 (third row) and Neq0568 (fourth row).**
(PDF)

## Author Contributions

**Conceptualization:** Geraldo Rodrigues Sartori, Carlos A. Montanari, Charles A. Laughton.

**Data curation:** Geraldo Rodrigues Sartori.

**Funding acquisition:** Andrei Leitão, Carlos A. Montanari, Charles A. Laughton.

**Investigation:** Geraldo Rodrigues Sartori.

**Methodology:** Charles A. Laughton.

**Project administration:** Carlos A. Montanari.

**Supervision:** Andrei Leitão, Carlos A. Montanari, Charles A. Laughton.

**Writing – original draft:** Geraldo Rodrigues Sartori, Charles A. Laughton.

**Writing – review & editing:** Andrei Leitão, Carlos A. Montanari.

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
