## [Decision Letter · Decision Letter 0]

11 Sep 2019

PONE-D-19-23070

Ligand-induced Conformational Selection Predicts the Selectivity of Cysteine Protease Inhibitors

PLOS ONE

Dear Dr. Laughton,

Thank you for submitting your manuscript to PLOS ONE. After careful consideration, we feel that it has merit but does not fully meet PLOS ONE’s publication criteria as it currently stands. Therefore, we invite you to submit a revised version of the manuscript that addresses the points raised during the review process.

This manuscript is on the border of a minor vs major revision.

Reviewer #2's points are fairly minor and should be addressed readily.

Reviewer #1 has three questions about the methodology used and one about the strength of a conclusion. These need to be addressed. It is possible they may be addressable without additional calculations, but additional calculations might be needed.

We would appreciate receiving your revised manuscript by Oct 26 2019 11:59PM. To enhance the reproducibility of your results, we recommend that if applicable you deposit your laboratory protocols in protocols.io, where a protocol can be assigned its own identifier (DOI) such that it can be cited independently in the future. For instructions see: http://journals.plos.org/plosone/s/submission-guidelines#loc-laboratory-protocols

We look forward to receiving your revised manuscript.

Kind regards,

Freddie Salsbury , Jr, PhD

Academic Editor

PLOS ONE

Journal Requirements:

2. We noted that several of your references did not auto populate and instead the manuscript contains the following  "Error! Reference source not found", please replace this with the appropriate reference during your next revision.

Reviewers' comments:

Reviewer's Responses to Questions

**Comments to the Author**

1. Is the manuscript technically sound, and do the data support the conclusions?

Reviewer #1: Yes

Reviewer #2: Yes

2. Has the statistical analysis been performed appropriately and rigorously? 

Reviewer #1: Yes

Reviewer #2: Yes

3. Have the authors made all data underlying the findings in their manuscript fully available?

Reviewer #1: Yes

Reviewer #2: No

4. Is the manuscript presented in an intelligible fashion and written in standard English?

Reviewer #1: Yes

Reviewer #2: Yes

5. Review Comments to the Author

Reviewer #1: Dear Editor, dear authors,

the submitted manuscript “Ligand-induced Conformational Selection Predicts the Selectivity of Cysteine Protease Inhibitors“ describes a method to identify potent inhibitors for a cysteine protease and to predict the selectivity of the respective inhibitor. Rodrigues Sartori and colleagues nicely show that potent nitril inhibitors typically bind to their target protease via conformational selection.

Overall the study is conclusive, but some minor issues need to be addressed before publication:

(1) The authors mention that ICK is a weak cathepsin L inhibitor because it shows a high ligand RMSD during their simulations. High RMSD means the ligand tends to leave the active site more easily. Can this RMSD also be used to access / categorize the potency of cysteine protease inhibitors? Why is it not included in the current decision tree? To me this would seem to be a straight forward approach.

(2) Nitrile inhibitors are covalent-reversible inhibitors. I would assume that potency of such an inhibitor correlates with the stability of the covalent complex. The stability of the covalent complex is not only defined by the energy of the cross-link formation, but also by the enzyme – inhibitor environment. Can the authors comment on that? How is this included in the calculations?

(3) Using the method described in the manuscript, is it possible to compare different classes of inhibitors? The energy of covalent bond formation might be different. How can this be included in the calculations?

(4) On page 11, lines 410 – 417 the authors conclude that inhibitors binding to cysteine proteases via conformational selection are in general stronger inhibitors as compared to those binding via induced fit. This conclusion is based on the observation that ICL is a weak cathepsin K inhibitor. In my opinion this is not enough evidence to draw such a general conclusion. It could as well be just a coincidence.

Overall the manuscript and the figures are clear and well-made and I believe that this report will be read with great interest by the scientific community.

Reviewer #2: This manuscript combines MD simulation, PCA analysis and decision tree to identify the activity of different compounds agsinst at papain-family proteins. The topic and protocol is interesting.

Just a few issues the author need to address:

1. The author needs to put the unit of the distance and RMSD into table 2's legend.

2. In line 317, it showed "Results are presented in Error! Reference source not found., and in more detail in S9-S12 Figs 318 in the supporting information.". Author needs to fix this error.

3. In order to compare, the rangle of X and Y axisis should be added in figure 2.

6. PLOS authors have the option to publish the peer review history of their article (what does this mean?). If published, this will include your full peer review and any attached files.

Reviewer #1: No

Reviewer #2: No

---

## [Author Response · Author response to Decision Letter 0]

1 Nov 2019

PONE-D-19-23070

Ligand-induced Conformational Selection Predicts the Selectivity of Cysteine Protease Inhibitors

PLOS ONE

Dear Dr. Salsbury,

We thank you and the reviewers for your guidance. Below we detail our responses to each of the issues raised, and hope that you find this satisfactory.

Regards,

Charlie Laughton

Journal Requirements:

Response: We have gone through the manuscript carefully, and hope now it fully matches up to PLOSOne’s style requirements.

2. We noted that several of your references did not auto populate and instead the manuscript contains the following "Error! Reference source not found", please replace this with the appropriate reference during your next revision.

Response: We have corrected some references; in our local version of the manuscript these format correctly (we use Mendeley) but will double-check this during the resubmission upload process.

Response: Datasets (molecular dynamics trajectory files) necessary to reproduce our study findings have been deposited in Zenodo. Here are the URLs and DOIs:

https://zenodo.org/record/3518308 10.5281/zenodo.3518308

https://zenodo.org/record/3522090 10.5281/zenodo.3522090

https://zenodo.org/record/3523307 10.5281/zenodo.3523307

https://zenodo.org/record/3523367 10.5281/zenodo.3523367

Reviewers' comments:

Reviewer's Responses to Questions

Comments to the Author

1. Is the manuscript technically sound, and do the data support the conclusions?

Reviewer #1: Yes

Reviewer #2: Yes

Response: no response required

2. Has the statistical analysis been performed appropriately and rigorously? 

Reviewer #1: Yes

Reviewer #2: Yes

 Response: no response required

3. Have the authors made all data underlying the findings in their manuscript fully available?

Reviewer #1: Yes

Reviewer #2: No

 Response: see above – datasets are now deposited at Zenodo

4. Is the manuscript presented in an intelligible fashion and written in standard English?

Reviewer #1: Yes

Reviewer #2: Yes

 Response: no response required

5. Review Comments to the Author

Reviewer #1: Dear Editor, dear authors,

the submitted manuscript “Ligand-induced Conformational Selection Predicts the Selectivity of Cysteine Protease Inhibitors“ describes a method to identify potent inhibitors for a cysteine protease and to predict the selectivity of the respective inhibitor. Rodrigues Sartori and colleagues nicely show that potent nitril inhibitors typically bind to their target protease via conformational selection.

Overall the study is conclusive, but some minor issues need to be addressed before publication:

(1) The authors mention that ICK is a weak cathepsin L inhibitor because it shows a high ligand RMSD during their simulations. High RMSD means the ligand tends to leave the active site more easily. Can this RMSD also be used to access / categorize the potency of cysteine protease inhibitors? Why is it not included in the current decision tree? To me this would seem to be a straight forward approach.

Response: We address this on page 13 of the manuscript in the section “Structural Analysis of Molecular Dynamics Simulations”. We show that while it is true that in some cases (e.g. ICK) high ligand RMSD and low affinity/potency correlate, they do not always do so. This is why we did not use this in the decision tree. We have added a sentence at the end of this section to clarify this.

(2) Nitrile inhibitors are covalent-reversible inhibitors. I would assume that potency of such an inhibitor correlates with the stability of the covalent complex. The stability of the covalent complex is not only defined by the energy of the cross-link formation, but also by the enzyme – inhibitor environment. Can the authors comment on that? How is this included in the calculations?

Response: This is a good point, we had not fully explained our approach. We have re-written part of the section “Binding Free Energy Calculations using the MMGBSA Method” to answer this question.

(3) Using the method described in the manuscript, is it possible to compare different classes of inhibitors? The energy of covalent bond formation might be different. How can this be included in the calculations?

Response: A good point, we have added a short discussion of this to the conclusions section; we suggest that a QM-derived term measuring the strength of the cross-link bond would be sufficient.

(4) On page 11, lines 410 – 417 the authors conclude that inhibitors binding to cysteine proteases via conformational selection are in general stronger inhibitors as compared to those binding via induced fit. This conclusion is based on the observation that ICL is a weak cathepsin K inhibitor. In my opinion this is not enough evidence to draw such a general conclusion. It could as well be just a coincidence.

Response: in fairness we only claimed the result “strengthened the hypothesis” rather than proved it, but we have altered the final part of this section to qualify our analysis further, in line with the referee’s comments.

Overall the manuscript and the figures are clear and well-made and I believe that this report will be read with great interest by the scientific community.

Reviewer #2: This manuscript combines MD simulation, PCA analysis and decision tree to identify the activity of different compounds agsinst at papain-family proteins. The topic and protocol is interesting.

Just a few issues the author need to address:

1. The author needs to put the unit of the distance and RMSD into table 2's legend.

Response: this has been done.

2. In line 317, it showed "Results are presented in Error! Reference source not found., and in more detail in S9-S12 Figs 318 in the supporting information.". Author needs to fix this error.

Response: this has been done.

3. In order to compare, the rangle of X and Y axisis should be added in figure 2.

 Response: We have amended the legend to this figure to say that all panels are at the same magnification (range of X and Y values), and what this is (40 Angstoms x 40 Angstroms).

---

## [Editor Report · Decision Letter 1]

20 Nov 2019

Ligand-induced Conformational Selection Predicts the Selectivity of Cysteine Protease Inhibitors

PONE-D-19-23070R1

Dear Dr. Laughton,

We are pleased to inform you that your manuscript has been judged scientifically suitable for publication and will be formally accepted for publication once it complies with all outstanding technical requirements.

With kind regards,

Freddie Salsbury , Jr, PhD

Academic Editor

PLOS ONE
---

## [Editor Report · Acceptance letter]

6 Dec 2019

PONE-D-19-23070R1 

Ligand-induced Conformational Selection Predicts the Selectivity of Cysteine Protease Inhibitors 

Dear Dr. Laughton:

I am pleased to inform you that your manuscript has been deemed suitable for publication in PLOS ONE. Congratulations! Your manuscript is now with our production department. 

With kind regards,

on behalf of

Dr. Freddie Salsbury , Jr 

Academic Editor

PLOS ONE